# Satellite-Borne Optical Remote Sensing Image Registration Based on Point Features

**DOI:** 10.3390/s21082695

**Published:** 2021-04-11

**Authors:** Xinan Hou, Quanxue Gao, Rong Wang, Xin Luo

**Affiliations:** 1School of Electronic Engineering, Xidian University, Xi’an 710071, China; xnhou@stu.xidian.edu.cn; 2School of Telecommunications Engineering, Xidian University, Xi’an 710071, China; qxgao@xidian.edu.cn; 3Yangtze Delta Region Institute (HuZhou), University of Electronic Science and Technology of China, Huzhou 313099, China; wrong2015@163.com; 4School of Resources and Environment, University of Electronic Science and Technology of China, Chengdu 611731, China

**Keywords:** optical remote sensing, image registration, point feature, rough matching, KNN-TAR

## Abstract

Since technologies in image fusion, image splicing, and target recognition have developed rapidly, as the basis of many image applications, the performance of image registration directly affects subsequent work. In this work, for rich features of satellite-borne optical imagery such as panchromatic and multispectral images, the Harris corner algorithm is combined with the scale invariant feature transform (SIFT) operator for feature point extraction. Our rough matching strategy uses the K-D (K-Dimensional) tree combined with the BBF (Best Bin First) method, and the similarity measure is the nearest neighbor/the second-nearest neighbor ratio. Finally, a triangle-area representation (TAR) algorithm is utilized to eliminate false matches in order to ensure registration accuracy. The performance of the proposed algorithm is compared with existing popular algorithms. The experimental results indicate that for visible light and multi-spectral satellite remote sensing images of different sizes and different sources, the proposed algorithm in this work is excellent in accuracy and efficiency.

## 1. Introduction

The specific objective of image registration is to find the geometric correspondence between different images that contain the same contents. It uses an accurate model to describe the internal relationship among pixels in two images and accurately match them together. Hence, a unified model is essential to find the commonalities of features in different images [1,2]. Image registration frequently appears in research works of scholars, and it possesses important value in multi-temporal and large-scale application scenarios. In the field of remote sensing, there are many types of satellite-borne optical image sensors, and their acquired remote sensing images are of different resolutions and bands. Though the accomplishment of registration, mosaicking and fusion among these images, more integral and abundant data can be generated, which can lay the foundation for subsequent work, such as change detection and scene expansion.

Recently, with the continuous development of sensor technology, researchers in the satellite remote sensing field have paid more attention to the study of high-speed and stable information transmission [3,4,5,6,7]. As sensor types become more and more diverse, remote sensing images with different spatial and spectral resolutions can be obtained using different satellite-borne platforms. At present, there are many types of observation satellites, including Landsat series and SPOT (Systeme Probatoire d’Observation de la Terre) series, as well as Chinese Gaofen (GF) series and ASTER (Advanced Spaceborne Thermal Emission and Reflection Radiometer). This work is mainly focused on the registration of satellite-borne optical remote sensing imagery, i.e., panchromatic and multispectral images.

The remainder of this article is organized as follows. The second section introduces the principle of the algorithm, including the Harris corner point algorithm, scale invariant feature transform (SIFT) algorithm, Best Bin First (BBF) algorithm and nearest/near neighbor ratio method. Then, the basic principle of the triangle-area representation (TAR) algorithm is described in detail, which is adopted to eliminate false matches in rough matching and realize fine registration. Meanwhile, the optimal affine transform parameters are obtained for the matched points. In the third section, the proposed algorithm in this work is evaluated by using the images from GF-1, GF-2 and ASTER. Finally, conclusions and discussions are given in the fourth section.

## 2. Remote Sensing Image Matching

This work uses the Harris algorithm to extract feature points from reference images and images to be registered, and then uses the SIFT operator to describe the feature points. The rough matching strategy is relied on the K-D (K-Dimensional) tree combined with the BBF algorithm, and the similarity measure is the first/second nearest neighbor ratio method. Finally, a TAR-based algorithm is used to eliminate false matches’ points in order to precisely match the image. Our algorithm procedure is demonstrated in Figure 1 as follows:

### 2.1. Remote Sensing Image Preprocessing

There are many factors that have an impact on optical remote sensing image quality. A major disadvantage is thermal noise or interference from other factors in the imaging processes. Another disadvantage is that the encoding mode of some image systems sacrifices grayscale representation to some degree, in order to achieve high compression ratios, which makes gray differences of pixels in adjacent regions smaller [8] and reduces the gradients of gray value between objects and backgrounds. Consequently, feature extraction for registration becomes difficult and it is necessary for these images to be filtered or enhanced, so as to improve their quality before deep processing [9,10,11].

For the purpose of improving the variation range of gray scale and contrast, linear stretching is adopted in this work. Generally, a linear stretch of 0.02 can achieve acceptable visual appearances in remote sensing images. That is, the distribution of an image histogram between 2% and 98% is linearly stretched to extend dynamic range of pixels to its whole gray space, so that the whole image has more abundant gray information. Figure 2 is an enhanced example of 0.02 linear stretching; the processed image has stronger contrast, better visual appearance, and is more beneficial in terms of subsequent feature extraction than the original one.

### 2.2. Feature Point Extraction

#### 2.2.1. Harris Feature Points Extraction

The Harris corner detection and extraction algorithm can build a rectangular window of a certain size to test every pixel in an image. The testing content is the average energy of the pixels in the window, which serves as the metric for judging whether a pixel is a corner point. Namely, when the average energy of the point in the window is greater than a preset threshold, it can be regarded as a feature point [12].

#### 2.2.2. SIFT Feature Point Description

Since it does not involve the construction of multi-scale space, the time complexity of the Harris operator is rather low. It has good robustness to illumination, scale, rotation and angle transformation [13], but its detection performance for smooth images is not satisfactory [14]. In spite of its higher time complexity, the SIFT operator can capture more feature points. Taking the main characteristics of optical remote sensing images into consideration, we use the Harris algorithm to extract feature points and apply the SIFT descriptor to describing the feature points in order to satisfy both accuracy and time requirements [15].

Generally, the SIFT algorithm consists of two parts: determining feature points of an image and describing feature points [16,17]. The process of image feature points determination is similar to the perception of point information by human vision. Usually, regardless of optical image resolution, human eyes can always distinguish valid features [18]. Hence, stable feature points of images are detected at different scales, and some information obtained in the detection process is utilized to construct a multi-dimensional description symbol to arrange the Harris feature points. In the term of feature point extraction, the SIFT operator is robust to scaling, rotation and transforms of images, and it is also resistant to external factors such as illumination and noise [19].

### 2.3. Rough Matching Strategy

#### 2.3.1. BBF Search Strategy

The K-D tree, is essentially a binary tree structure. In this search space, data are usually divided into a binary tree structure according to spatial positions, and then search processes are also carried out according to the search rules of binary trees. The kernel idea of the K-D tree is to divide data into left and right uniform structures from top to bottom in a two-dimensional space. As a result, all data are split into a left subtree and a right subtree according to their spatial position, and then the same operation is repeated on the subtrees to part them into smaller subtrees until the data cannot be subdivided. In the process of dividing, it is vital to maintain the data balance between the left and right subtrees as much as possible. Otherwise, search efficiency may decline. The Best Bin First (BBF) search algorithm is a search algorithm developed for K-D tree structures. It outperforms the K-D tree search algorithm in terms of processing high-dimensional features [20]. It pushes points that can be traced into a sequence, and sorts these points, according to their distances from a hyperplane. The closest point has the highest priority, and then all the points in the queue are traversed according to their priority until the queue becomes empty. In addition, BBF also imposes restrictions on its search time. That is, once its running time exceeds the pre-set time, the algorithm will directly output the current closest point as the result. Therefore, we chose the K-D tree to organize the feature points while using the BBF algorithm to search for the feature points in this work.

#### 2.3.2. Similarity Measure

The first/second nearest neighbor ratio method is chosen as the similarity metric in registration for reducing the complexity of calculation [21]. First of all, the point in an image to be registered that is closest to a search point in its reference image needs to be found, and their distance is denoted as *Dis*1. Then, it is necessary to find the next nearest point to the search point, and their distance is denoted as *Dis*2. Then, when comparing *Dis*1/*Dis*2 to a given threshold, and when the ratio is less than the threshold, the point pair can be considered to be a possible real matching pair.

### 2.4. Fine Matching Strategy

Since satellite-borne optical remote sensing images are acquired at high altitude, the view differences among images of the same target are slight. Changes between images only involve transforms such as translation, scaling, and rotation. Therefore, affine transform models can satisfy geometric transform requirements in our cases [22,23]. Therefore, an affine transform based on Triangle-area representation (TAR) is utilized in fine matching [24].

TAR is a framework in which features at every scale, i.e., edge lengths of a triangle, are normalized locally according to their scales. Among shape attributes at different scales, local normalization features are more distinct and can more accurately describe shapes in remote sensing images. Unlike some other matching methods that use a limited number of boundary points (such as corners or knee points), TAR is an exhaustive method for all boundary points.

TAR value is calculated from a triangle region formed by points on the shape boundary. Each contour point is represented by its coordinate (*x*, *y*), and a discrete parameter sequence (*x_n_*, *y_n_*) (*n* = 1, …, *N*) represents the *N* points obtained by resampling a shape. Then, the curvature of each point is represented using the TAR value defined below. For three sequential points, $\left(x_{n-t_{s}},y_{n-t_{s}}\right)$, $\left(x_{n},y_{n}\right)$ and $\left(x_{n+t_{s}},y_{n+t_{s}}\right)$, where *n* ∈ [1, *N*], *t_s_* ∈ [1, *T_s_*] represents arbitrary edge length of a triangle, and *T_s_* is the longest distance between any two sample points. Thus, the TAR value formed by these points can be expressed as:(1)$$TAR\left(n,t_{s}\right)=\frac{1}{2}\left|\begin{array}{ccc} x_{n-t_{s}} & y_{n-t_{s}} & 1\\ x_{n} & y_{n} & 1\\ x_{n+t_{s}} & y_{n+t_{s}} & 1 \end{array}\right|$$

Given that running counterclockwise, the TAR value is positive when the local contour denoted by three sample points is convex. The TAR value is negative when the contour is concave. And the TAR value is 0 when the contour is straight. Figure 3 depicts triangular regions at different positions of a closed contour.

The above figure is a closed hammer-shaped contour. Region 1 is a convex shape, and its TAR value is greater than 0. Region 2 is a concave shape, and its TAR value is less than 0. Region 3 is a straight line. The curvature function of the TAR value of discrete points can be rewritten as:(2)$$c\left(n\right)=\frac{{\dot{x}}_{n}{\ddot{y}}_{n}-{\ddot{x}}_{n}{\dot{y}}_{n}}{{\left({\dot{x}}_{n}^{2}+{\dot{y}}_{n}^{2}\right)}^{\frac{3}{2}}}=\frac{TAR\left(n,n+1\right)}{{\left(d_{s_{n}}\right)}^{3}},$$
where *TAR*(*n*, *n* + 1) is the TAR value at *t_s_* = 1, and $d_{s_{n}}=\sqrt{{\dot{x}}_{n}^{2}+{\dot{y}}_{n}^{2}}$ corresponds to the first edge length of a triangle, that is, the distance between the first and second vectors of the triangle formed by points (*x_n_*, *y_n_*), (*x_n_*_+1_, *y_n_*_+1_), and (*x_n_*_+2_, *y_n_*_+2_). This equation clearly expresses the relationship between the curvature and TAR value of a shape. It is known that zero crossings of a curvature function are invariant under a general affine transformation [26], and points with non-zero curvature are also not invariant to the affine transformation [27]. Thus, considering the contour sequences *x_n_* and *y_n_* of a two-dimensional shape, if an affine transform is performed, the relationship between the original sequences and the transformed sequences is:(3)$$\left[\begin{array}{c} {\hat{x}}_{n}\\ {\hat{y}}_{n} \end{array}\right]=\left[\begin{array}{cc} a_{1} & a_{2}\\ a_{3} & a_{4} \end{array}\right]\left[\begin{array}{c} x_{n}\\ y_{n} \end{array}\right]+\left[\begin{array}{c} b_{1}\\ b_{2} \end{array}\right],$$
where ${\hat{x}}_{n}$ and ${\hat{y}}_{n}$ are the transformed sequences after the affine transform, *b*_1_ and *b*_2_ are translation parameters, and *a*_1_, *a*_2_, *a*_3_ and *a*_4_ are scaling and rotation factors. The influence of translation parameters is easily eliminated by normalizing the shape boundary corresponding to a centroid. This normalization is accomplished by subtracting the average value from each boundary sequence. Substituting Equation (3) into Equation (1), we can obtain:(4)$$T\hat{A}R\left(n,t_{s}\right)=\left(a_{1}a_{4}-a_{2}a_{3}\right)TAR\left(n,t_{s}\right)$$
where $T\hat{A}R$ is *TAR* value after the affine transformation. It is obvious that $T\hat{A}R$ is invariant for affine transformation.

### 2.5. Elimilation of Fales Matches

The difficulty of feature points matching lies in noise, intensive affine transformation, etc. For instance, some feature points are shifted from their original positions and become abnormal points in an image. Nowadays, there are many point matching algorithms, most of which are based on the similarity of local features, spatial relationships, or both. In some existing algorithms, affine invariant operators are utilized to detect whether a matching point is an abnormal point, through global information [28]. For example, the RANSAC (Random Sample Consensus) algorithm establishes a model for the correspondence between point pairs to estimate transform parameters. If false matches are not more than 50%, the algorithm can eliminate them effectively [29]. In this work, we use the affine invariance of TAR to eliminate false matches. The procedure consists of three steps: constructing KNN-TAR (K-Nearest Neighbor-Triangle-Area Representation) operators [30], processing candidate outliers and removing false matches.

Most of the outliers can be found by KNN-TAR, but a few outliers have the same nearest neighbors. The removal of such outliers is very important, and it directly effects the registration performance of the proposed algorithm. The outlier removal in this study includes three parts: the KNN-TAR descriptor, the process of the candidate outliers, and the removal of the remaining outliers. That is:Constructing KNN-TAR operators. Supposing that the nearest neighbors of the outliers have more structural dissimilarity, the TAR value is used to construct an affine invariant variable, which is calculated by the K nearest neighbor (KNN) in order to find outliers.Dealing with candidate outliers. Whether the suspected outliers sifted by KNN-TAR are real false matches is determined by the local structure of the single matching pair and the global transform error.Removing false matches. Adjust the parameter setting of KNN-TAR, so as to eliminate the outliers with the same KNN.

## 3. Experimental Results

The setup of the hardware environment requires an Intel core i5-4570 processor at 3.20 Hz, with 4.00 GB RAM. The operating system is 64-bit Windows 7, and our programming software is MATLAB R2014a. Here, we compare the proposed algorithm with other existing algorithms using four pairs of satellite-borne optical remote sensing images.

The first pair of experimental images is acquired by the GF-1 Panchromatic and Multispectral sensor (PMS) at Mao County, which is a multispectral image with 3000 × 1012 pixels. The preprocessed images are shown in Figure 4.

Figure 5 and Figure 6 display the extracted feature points and the rough matching result for the GF-1 image pair, respectively. It can be seen that our proposed algorithm in this work can detect enough point features.

By rough matching, 251 pairs of points are obtained. The straight lines in Figure 6 connect the corresponding points in the two images. It is obvious that there are some crossing lines between the two images, that is, there are obvious false matches. After the RANSAC algorithm eliminated one point pair, the obtained affine transform matrix for registration is presented in Equation (5).
(5)$$H_{RANSAC}=\left[\begin{array}{ccc} 1.0292 & -0.0006 & -363.8828\\ -0.0003 & 1.0308 & 50.3609\\ 0 & 0 & 1 \end{array}\right]$$

After 16 point pairs were eliminated by the proposed KNN-TAR algorithm, the obtained affine transform matrix for registration is given in Equation (6).
(6)$$H_{TAR}=\left[\begin{array}{ccc} 1.0293 & -0.0007 & -363.9619\\ -0.0004 & 1.0308 & 50.3757\\ 0 & 0 & 1 \end{array}\right],$$

Figure 7 exhibits the GF-1 image pairs after eliminating false matches using the two methods. The registration results for GF-1 are listed in Table 1. The RMSE (Root Mean Square Error) of our algorithm is 0.8619 pixels, which is better than that of the RANSAC algorithm.

Moreover, according to the obtained transformation parameters, the bilinear interpolation method is utilized to realize image mosaicking. The final stitched GF-1 image is shown in Figure 8.

The second pair of experimental images was acquired by the GF-2 PMS sensor at Mao County, which is a multispectral image with 2400 × 1800 pixels. The preprocessed and stitched GF-2 images are shown in Figure 9. The registration results for GF-2 are listed in Table 2. The RMSE of our algorithm is 5.7423 pixels, which is also better than that of RANSAC.

The third pair of experimental images are acquired by the visible/near infrared part of the ASTER sensor, with spatial resolutions of 15 m. The preprocessed and stitched ASTER images are shown in Figure 10 and they are pseudo-color synthetic images. The registration results for ASTER are listed in Table 3. The RMSE of our algorithm is 0.5362 pixels, which is also better than that of the RANSAC algorithm.

The fourth pair of experimental images is from GF-1 and GF-2. The preprocessed and stitched images are shown in Figure 11. The registration results for ASTER are listed in Table 4. The RMSE of our algorithm is 0.5362 pixels, which is also better than that of the RANSAC algorithm.

From the above experimental results, it is found that as the number of rough matching points increases, there are slight increases in the matching time of the KNN-TAR algorithm compared to that of the RANSAC algorithm. However, as for satellite-borne optical remote sensing images from different sources, the proposed algorithm in this work can eliminate false matches and realize accurate registration. Through comparative experiments, it can be proven that its registration accuracy is better than that of the RANSAC algorithm.

Moreover, we also compare the proposed method with three popular image registration methods, SIFT, SURF (Speed-Up Robust Features) and ORB (Oriented FAST and Rotated BRIEF). They also combined the BBF with RANSAC algorithm in feature point matching, and the SIFT algorithm still uses the Harris operator for feature extraction instead of global matching in the original version. The comparison results are also presented in Table 5. As revealed in Table 5, the proposed registration method outperforms the other methods for GF-1, ASTER and Multi-view images. Even for the GF-2 image, its registration result is also acceptable.

## 4. Conclusions

Because of small view differences of satellite remote sensing images, a registration method based on point features is designed in this study. The Harris operator, with its fast detection speed, is chosen to extract image features, and the SIFT operator is used to describe the features in order to ensure accuracy. After that, the BBF algorithm combined with the first/second-nearest neighbor method is adopted to realize rough matching of feature points. Then, a TAR method is introduced into false match elimination in order to enhance matching accuracy. The experimental results indicate that the method used in this work has better registration accuracy compared with RANSAC and some other existing registration algorithms. However, due to the combination of different optimization methods, the proposed algorithm has no significant advantages in time efficiency. With the increase in image resolution and size, registration time will increase. Therefore, other effective optimization algorithms may be utilized to accelerate parameter fitting processes to improve the speed of registration. Our proposed strategy is only effective in two-dimensional space, and cannot perform well for images with many view differences. In the future, it can be modified to adapt to three-dimensional space.

## Figures and Tables

**Figure 1 sensors-21-02695-f001:**
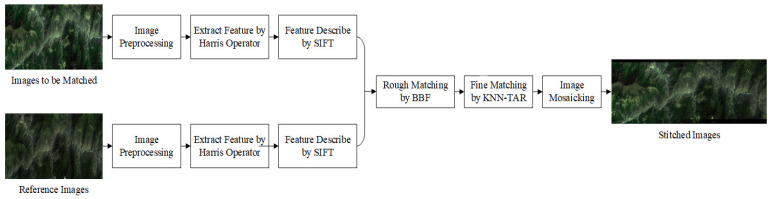
The proposed image registration algorithm flow chart.

**Figure 2 sensors-21-02695-f002:**
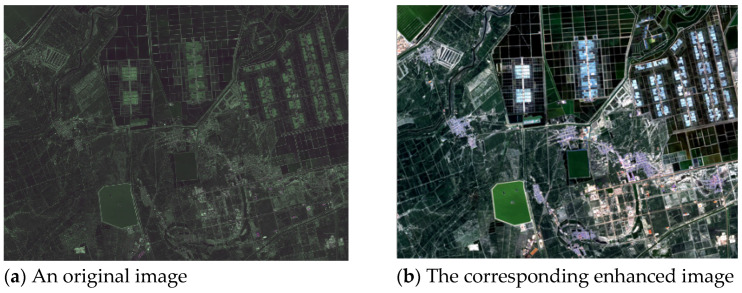
Stretching of a pseudo-color image synthetized from a GF-1 (Gaofen-1) multispectral image, acquired at 38° N and 117.7° E on 9 September 2014: (**a**) An original image; (**b**) the corresponding enhanced image.

**Figure 3 sensors-21-02695-f003:**
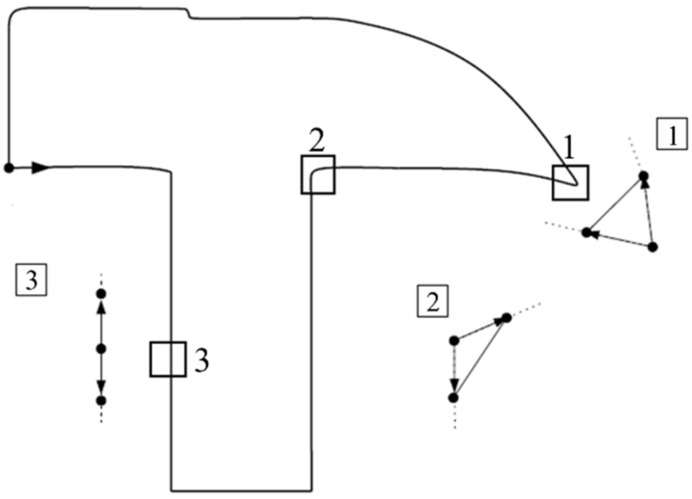
Triangular regions of a closed contour [25].

**Figure 4 sensors-21-02695-f004:**
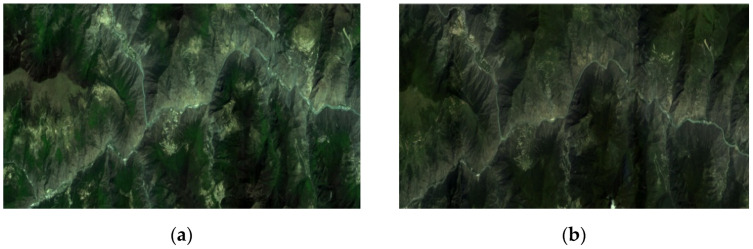
A GF-1 image pair: (**a**) acquired on 19 February 2015; (**b**) acquired on 11 May 2015.

**Figure 5 sensors-21-02695-f005:**
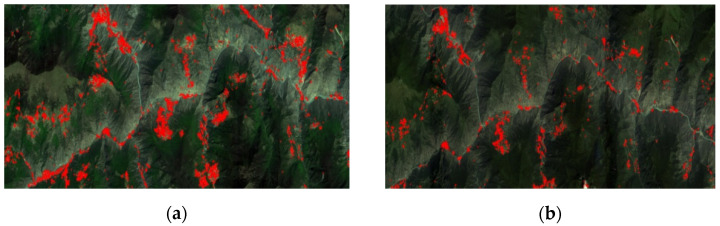
The extracted feature points in the GF-1 image pair: (**a**) acquired on 19 February 2015; (**b**) acquired on 11 May 2015.

**Figure 6 sensors-21-02695-f006:**
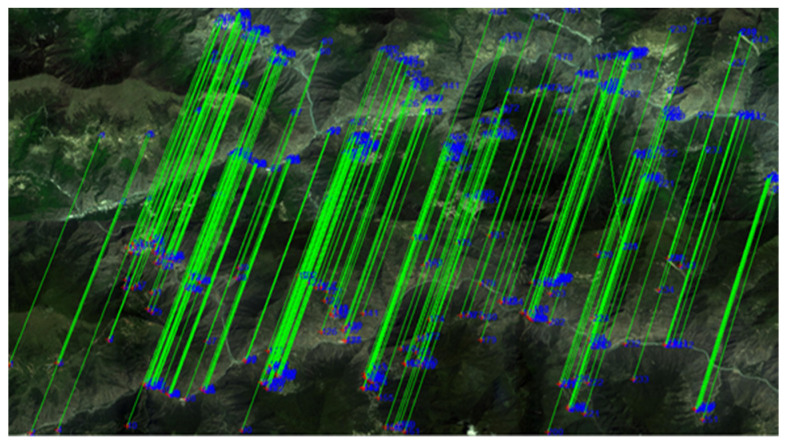
The rough matching result of the GF-1 image pair.

**Figure 7 sensors-21-02695-f007:**
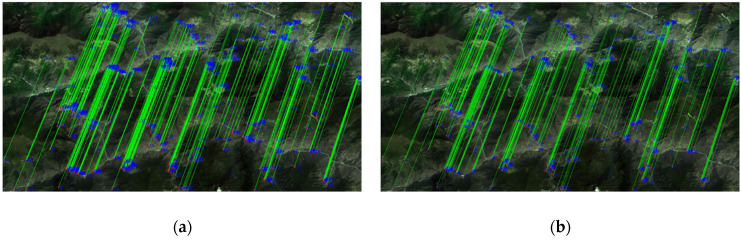
The GF-1 image pairs after eliminating false matches: (**a**) Finely matched by the RANSAC (Random Sample Consensus) algorithm; (**b**) finely matched by the KNN-TAR (K-Nearest Neighbor-Triangle-Area Representation) algorithm.

**Figure 8 sensors-21-02695-f008:**
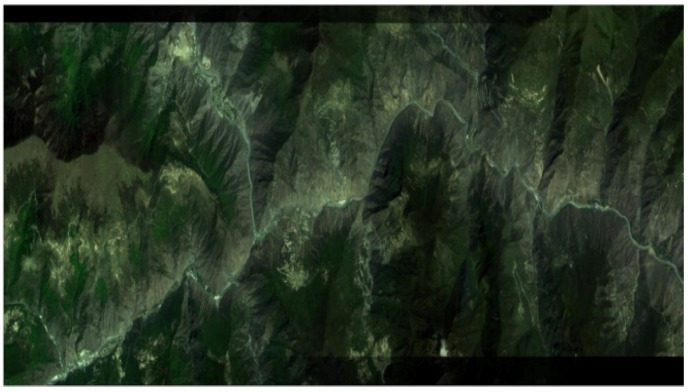
The stitched GF-1 image.

**Figure 9 sensors-21-02695-f009:**
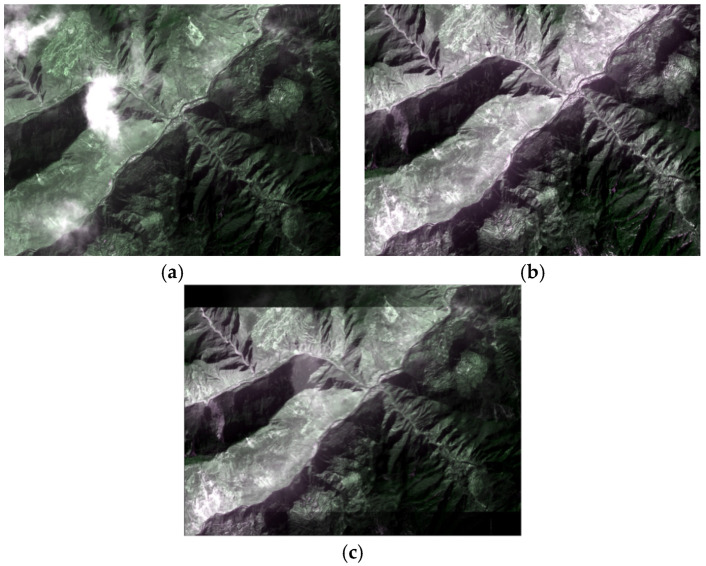
A GF-2 image pair: (**a**) Acquired on 19 February 2015; (**b**) acquired on 24 February 2015; (**c**) the stitched GF-2 image.

**Figure 10 sensors-21-02695-f010:**
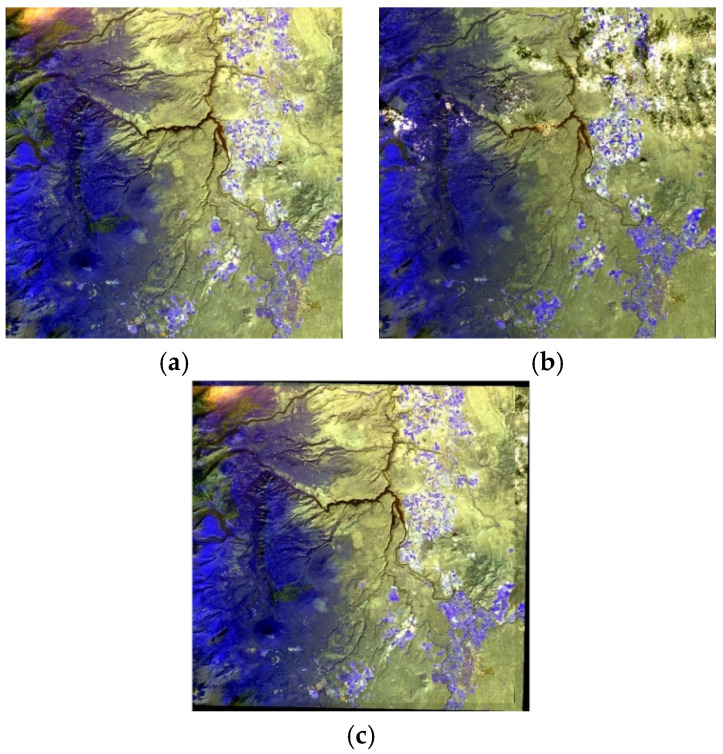
An ASTER (Advanced Spaceborne Thermal Emission and Reflection Radiometer) image pair: (**a**) Acquired on 15 August 2020; (**b**) acquired on 20 October 2020; (**c**) the stitched ASTER image.

**Figure 11 sensors-21-02695-f011:**
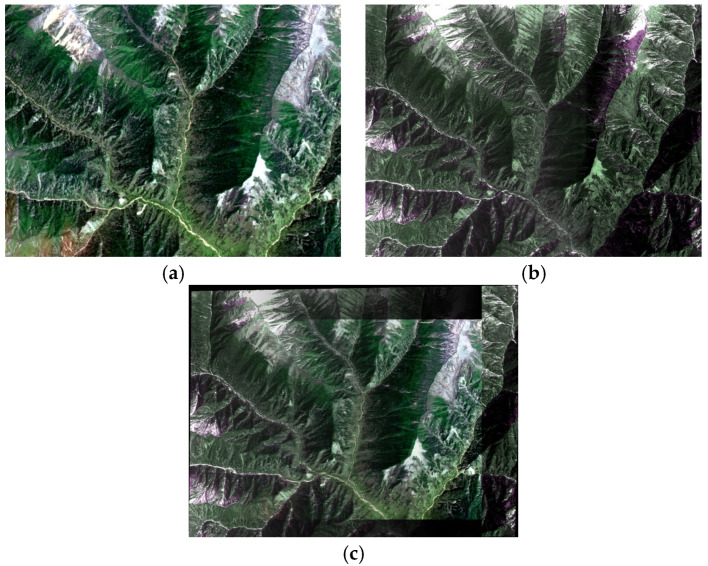
A multi-source image pair: (**a**) Acquired by GF-1 on 19 February 2015 (800 × 600); (**b**) acquired by GF-2 on 24 February 2015 (2400 × 1800); (**c**) the stitched multi-source image.

**Table 1 sensors-21-02695-t001:** Comparison of the KNN-TAR and RANSAC algorithms for GF-1 images.

Match Methods	Number of Pairs by Rough Matching	Number of Pairs by Fine Matching	Time (ms)	RMSE (Pixels)
RANSAC	251	250	44	0.8818
KNN-TAR		235	51	0.8619

**Table 2 sensors-21-02695-t002:** Comparison of the KNN-TAR and RANSAC algorithms for GF-2 images.

Match Methods	Number of Pairs by Rough Matching	Number of Pairs by Fine Matching	Time (ms)	RMSE (Pixels)
RANSAC	1150	1150	96	5.8743
KNN-TAR		1135	123	5.7423

**Table 3 sensors-21-02695-t003:** Comparison of TAR and RANSAC algorithms for ASTER images.

Match Methods	Number of Pairs by Rough Matching	Number of Pairs by Fine Matching	Time (ms)	RMSE (Pixels)
RANSAC	111	111	20	0.5666
KNN-TAR		103	23	0.5362

**Table 4 sensors-21-02695-t004:** Comparison of the TAR and RANSAC algorithms for multi-source image images.

Match Methods	Number of Pairs by Rough Matching	Number of Pairs by Fine Matching	Time (ms)	RMSE (Pixels)
RANSAC	26	25	33	3.0044
KNN-TAR		24	28	2.9001

**Table 5 sensors-21-02695-t005:** Registration performance comparison of different methods.

Image Sensors	SIFT		SURF [31]		ORB [32]		Proposed Algorithm	
	Matched Point Pairs	RMSE (Pixels)	Matched Point Pairs	RMSE (Pixels)	Matched Point Pairs	RMSE (Pixels)	Matched Point Pairs	RMSE (Pixels)
GF-1	250	0.8818	114	1.0976	41	1.2530	235	0.8619
GF-2	1150	5.8743	1221	5.5924	63	1.6358	1135	5.8423
ASTER	111	0.5666	64	0.7330	17	1.3132	103	0.5362
GF-1 & GF-2	23	3.0044	70	5.2689	47	3.6736	24	2.9001

## Data Availability

The download website of the ASTER data images used in this work is https://e4ftl01.cr.usgs.gov/, accessed on 10 April 2021. GF-1 and GF-2 images are available at the following link: http://www.cresda.com/CN/sjfw/zxsj/index.shtml, accessed on 10 April 2021.

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
