# Peer review of "Satellite-Borne Optical Remote Sensing Image Registration Based on Point Features"

_sensors, 2021, doi:10.3390/s21082695_

Round 1
Reviewer 1 Report
This manuscript studied the registration model of visible remote sensing images. The authors didn’t propose a new registration method, but combined several existing technologies in the different steps of image registration. Anyway, the result is pleased. However, some other problems in the manuscript are still concerned in the following:
1. More backgrounds should be exposed in the introduction.
2. The organization of this manuscript should be added to the end of the introduction.
3. Could the authors show the flow chart of the method used in the studied?
4. Could the authors show more results of registration in the experiments? They just showed one case in the manuscript. In my opinion, it’s not convincing.
5. Could the authors compare with more registration methods in the experiments?
6. The quality of figures should be improved.
7. Most references are old. More recent works on remote sensing image registration are suggested, such as “DOI: 10.1016/j.jpdc.2021.02.014”, “DOI: 10.1016/j.isprsjprs.2019.03.002”…
8. Some references missed page. Please check them.
Author Response
Response to Reviewer 1 Comments
- More backgrounds should be exposed in the introduction.
A: The related content has been added to the introduction in the current modified version.
- The organization of this manuscript should be added to the end of the introduction.
A: The related content has been added to the introduction in the current modified version.
- Could the authors show the flow chart of the method used in the studied?
A: An Algorithm flowchart has been added into the current modified version..
- Could the authors show more results of registration in the experiments? They just showed one case in the manuscript. In my opinion, it’s not convincing.
A: Three image pairs of GF-2, GF-1/GF-2 and ASTER have been added to the experimental results.
- Could the authors compare with more registration methods in the experiments?
A: We have used four algorithms to compare the registration results for four pairs of test images.
- The quality of figures should be improved.
A: This problem has been resolved.
- Most references are old. More recent works on remote sensing image registration are suggested, such as “DOI: 10.1016/j.jpdc.2021.02.014”, “DOI: 10.1016/j.isprsjprs.2019.03.002”…
A: This problem has been resolved.
- Some references missed page. Please check them.
A: This problem has been resolved.
Reviewer 2 Report
*) The paper is very interesting and the topic is treated with a scientifically satisfactory methodology. From my point of view, I noticed that the image enhancement was treated with techniques that do not take into account any uncertainties and / or inaccuracies contained in the images. Notwithstanding that such a study is outside the scope of this work (but which can be taken into consideration for any future developments of the research in progress) I advise the authors to include in the text at least one sentence that highlights this possibility by putting the following relevant works in the bibliography :
doi: 10.1007/s40815-020-01030-5
doi: 10.1007/s11042-020-08699-8
*) Many acronyms have been used in the text. Perhaps it would be useful to insert a table that lists them. This would help the reader to read the paper.
*) The conclusions appear rather poor and would require additional comments on the results obtained. Also, if possible, I advise authors to provide at least one line of future research to make the paper more interesting to the scientific community.
Author Response
Response to Reviewer 2 Comments
*) The paper is very interesting and the topic is treated with a scientifically satisfactory methodology. From my point of view, I noticed that the image enhancement was treated with techniques that do not take into account any uncertainties and / or inaccuracies contained in the images. Notwithstanding that such a study is outside the scope of this work (but which can be taken into consideration for any future developments of the research in progress) I advise the authors to include in the text at least one sentence that highlights this possibility by putting the following relevant works in the bibliography :
doi: 10.1007/s40815-020-01030-5
doi: 10.1007/s11042-020-08699-8
A: This problem has been resolved.
*) Many acronyms have been used in the text. Perhaps it would be useful to insert a table that lists them. This would help the reader to read the paper.
A: The full names of all abbreviations in this article are given in the current modified version.
*) The conclusions appear rather poor and would require additional comments on the results obtained. Also, if possible, I advise authors to provide at least one line of future research to make the paper more interesting to the scientific community.
A: The conclusion part has been revised accordingly.
We would like to express our great appreciation to you and reviewers for comments on our paper. Looking forward to hearing from you.
Thank you and best regards.
Round 2
Reviewer 1 Report
All my comments have been solved.
